# Maternal and Fetal Bile Acid Homeostasis Regulated by Sulfated Progesterone Metabolites through FXR Signaling Pathway in a Pregnant Sow Model

**DOI:** 10.3390/ijms23126496

**Published:** 2022-06-10

**Authors:** Peng Wang, Peiqiang Yuan, Sen Lin, Heju Zhong, Xiaoling Zhang, Yong Zhuo, Jian Li, Lianqiang Che, Bin Feng, Yan Lin, Shengyu Xu, De Wu, Douglas G Burrin, Zhengfeng Fang

**Affiliations:** 1Key Laboratory for Animal Disease Resistance Nutrition of the Ministry of Education, Animal Nutrition Institute, Sichuan Agricultural University, Chengdu 611130, China; wangpeng@haut.edu.cn (P.W.); yuanpq2017@foxmail.com (P.Y.); zhongheju@foxmail.com (H.Z.); zhangxl@sicau.edu.cn (X.Z.); ayb1983@163.com (Y.Z.); lijian522@hotmail.com (J.L.); clianqiang@hotmail.com (L.C.); fengbin@sicau.edu.cn (B.F.); able588@163.com (Y.L.); shengyu_x@hotmail.com (S.X.); wude@sicau.edu.cn (D.W.); 2College of Biology Engineering, Henan University of Technology, Zhengzhou 450000, China; 3Sericultural & Agri-Food Research Institute, Guangdong Academy of Agricultural Sciences, Guangzhou 510000, China; linsen@gdaas.cn; 4Key Laboratory for Food Science and Human Health, College of Food Science, Sichuan Agricultural University, Ya’an 625014, China; 5USDA/ARS Children’s Nutrition Research Center, Section of Gastroenterology, Hepatology and Nutrition, Department of Pediatrics, Baylor College of Medicine, Houston, TX 77030, USA; doug.burrin@usda.gov

**Keywords:** bile acid homeostasis, pregnancy, sulfated progesterone metabolites, FXR, sows

## Abstract

Abnormally elevated circulating bile acids (BA) during pregnancy endanger fetal survival and offspring health; however, the pathology and underlying mechanisms are poorly understood. A total of nineteen pregnant sows were randomly assigned to day 60 of gestation, day 90 of gestation (G60, G90), and the farrowing day (L0), to investigate the intercorrelation of reproductive hormone, including estradiol, progesterone and sulfated progesterone metabolites (PMSs), and BA in the peripheral blood of mother and fetuses during pregnancy. All data were analyzed by Student’s *t*-test or one-way ANOVA of GraphPad Prism and further compared by using the Student–Newman–Keuls test. Correlation analysis was also carried out using the CORR procedure of SAS to study the relationship between PMSs and BA levels in both maternal and fetal serum at G60, G90, and L0. Allopregnanolone sulphate (PM4S) and epiallopregnanolone sulphate (PM5S) were firstly identified in the maternal and fetal peripheral blood of pregnant sows by using newly developed ultraperformance liquid chromatography-tandem mass spectrometry (UPLC-MS/MS) methods. Correlation analysis showed that pregnancy-associated maternal BA homeostasis was correlated with maternal serum PM4S levels, whereas fetal BA homeostasis was correlated with fetal serum PM5S levels. The antagonist activity role of PM5S on farnesoid X receptor (FXR)-mediated BA homeostasis and fibroblast growth factor 19 (FGF19) were confirmed in the PM5S and FXR activator co-treated pig primary hepatocytes model, and the antagonist role of PM4S on FXR-mediated BA homeostasis and FGF19 were also identified in the PM4S-treated pig primary hepatocytes model. Together with the high relative expression of *FGF19* in pig hepatocytes, the pregnant sow is a promising animal model to investigate the pathogenesis of cholestasis during pregnancy.

## 1. Introduction

Pregnancy is associated with the significant physiological and metabolic changes of a mother’s organism to accommodate the changing needs of the growing fetus, and a notable feature is progressive increase in serum bile acid (BA) levels [1,2]. In most women there is a moderate increase within normal range, yet a subset of women with serum BA level above reference ranges (≥10 μmol/L) develop intrahepatic cholestasis of pregnancy (ICP) [3]. The incidence of ICP is much higher in twin than in single pregnancies (6.7–20.9% vs. 1.3–4.7%) [4,5]. The elevated maternal serum BA levels leads to the accumulation of BA in fetal blood circulation, and thus increases the risk of fetal distress, unexplained stillbirth, and perinatal mortality [6,7,8,9]. 

Evidence that disruption of BA homeostasis causes adverse fetal outcomes suggests that alleviating maternal elevated serum BA may improve fetal outcomes [10,11]. However, the complex mechanisms that explain the increased maternal peripheral serum BA during pregnancy are poorly understood and limited intervention strategies are available. Estrogens and progesterone likely play an important role in the pathogenesis of disrupted BA homeostasis. Previous studies have demonstrated that estrogen, such as estradio (estradiol-17β-glucuronide), is involved in regulating BA metabolism through the estrogen receptor, such as estrogen receptor α [1,12]. In contrast to estradiol, there was no report about the direct interaction of progesterone and BA metabolism. Recently, studies of pregnant women and female mice have identified that epiallopregnanolone sulphate (PM5S) reduced BA secretion by inhibiting farnesoid X receptor (FXR) signaling [13,14]. Thus, further study is warranted to explore the potential role of estradiol and sulfated progesterone metabolites (PMSs) in regulating BA metabolism during pregnancy.

There are many similarities between pregnant humans and sows regarding anatomy, genetics and physiology, gestation length, multiple pregnancies, and the BA composition and placental secretion of estradiol and progesterone [15,16,17]. Consistent with the observations in pregnant women and pregnant mice [1,18], the disruption of BA homeostasis, which is accompanied with decreased BA secretion into the gut and decreased levels of serum fibroblast growth factor 19 (FGF19), also occurred in pregnant sows [19]. FXR, one of the main BA receptors, regulates BA homeostasis by sensing the BA profile and levels [20]. It is complex to know whether the decreased FGF19, the main downstream of FXR, was the cause or result of dysregulated BA. Therefore, the aim of the current study was to reveal the relationship between PMSs and BA in the peripheral blood of mother and fetuses at G60, G90, and L0, and the regulation mechanism of PMSs to the BA metabolism through the FXR signaling pathway.

## 2. Results

### 2.1. Sulfated Progesterone Metabolism in Mother during Pregnancy

Based on the association of reproductive hormones and its metabolites with BA metabolism [13,21,22], we determined the dynamic change of estradiol, progesterone, and PMSs between mother and fetus. In contrast to the gradual increase (*p* < 0.01) in estrogen levels in maternal and fetal serum (Figure 1 and Appendix A), progesterone levels decreased (*p* < 0.01) from G60 to L0 (Figure 2a). Notably, progesterone metabolites PM4S and PM5S showed a trend of change similar to that of total BA (TBA) levels, increasing (*p* < 0.05) from G60 to G90 then decreasing (*p* < 0.05) from G90 to L0 (Figure 2a). Consequently, the sum concentration of PM4S and PM5S, especially PM4S, was much higher than progesterone and became the dominant form of progesterone metabolites. Correlation analysis showed that the levels of PM4S (*p* < 0.01) and PM5S (*p* < 0.10) were positively correlated with maternal serum TBA levels at G60, G90, and L0, whereas no correlation was found between progesterone and maternal TBA levels (Figure 2b–d and Appendix A). Consistent with pregnant women [23], the placenta of the pregnant sow is also able to synthesize progesterone [17]. The relative expression levels of placental sulfotransferase 2A1 (*SULT2A1*), sulphonating hydroxysteroids, such as pregnenolone [24], showed a similar trend with the change of PM4S and PM5S levels (Figure 3), which might account for the dynamic change of PM4S and PM5S during pregnancy, especially when the relative expression levels of *SULT2A1* in maternal livers was not different between G60 and G90 [19]. These results suggested that the dysregulation of maternal BA homeostasis was associated with the systemic accumulation of PMSs.

### 2.2. Sulfated Progesterone Metabolism in Fetuses during Pregnancy

In contrast to mothers, the dynamic change of serum progesterone and its sulfated metabolites in fetuses were different. The levels of progesterone, PM4S and PM5S, were significantly higher (*p* < 0.05) at L0 than at G60, while the levels of PM5S was higher (*p* < 0.05) at G90 than at G60 and the levels of progesterone and PM5S were also higher (*p* < 0.05) at L0 than at G90 (Figure 4a). Consequently, the sum of PM4S and PM5S was not higher than progesterone. Unlike the dominant forms of PM4S in maternal serum, the levels of PM5S in fetal serum was much higher than PM4S and were ~1.5 fold (51.77 vs. 33.82 nmol/L) and ~5.5 fold (123.49 vs. 22.20 nmol/L) higher than its counterpart in mothers at G90 and L0, respectively, whereas PM4S was continuously far lower than its counterparts in mothers at G60 (14.94 vs. 393.95 nmol/L), G90 (31.74 vs. 800.97 nmol/L), and L0 (48.76 vs. 280.43 nmol/L), respectively. Consistent with the result in mothers, correlation analysis showed that the levels of PM4S (*p* < 0.10) and PM5S (*p* < 0.01) were positively correlated with fetal serum TBA levels, respectively, and no significant correlation (*p* > 0.05) was found between progesterone and fetal TBA levels (Figure 4b–d).

### 2.3. PM4S Inhibits FXR-Mediated BA Metabolism

In consideration of the relationship between PM4S with TBA in mother and fetuses during pregnancy, we further investigate the role of PM4S on hepatocyte BA metabolism. Primary pig hepatocytes were treated with increasing concentrations of PM4S. Consistent with the positive relationship between PM4S and TBA in mothers, PM4S significantly increased the TBA levels in culture medium (Figure 5). PM4S also induced a gradually decreased the expression of genes involved in BA uptake (*NTCP*, *OATP1A2*), BA secretion (*BSEP* and *MRP2*), BA detoxification (*SULT2A1*), and BA efflux into the blood circulation (*OSTβ*), though genes involved in BA synthesis (*CYP7A1*, *CYP8B1*) were not significantly altered (Figure 6a–h). Multidrug resistance protein 2 (*MDR3*) mediated the secretion of phosphatidylcholine, which was also an important cause of cholestasis [25]. PM4S decreased the relative expression levels of *MDR3* in dose dependent manner (Figure 6i).

In consideration of the critical role of FXR in BA homeostasis, we further investigated the changes of FXR signaling pathways after PM4S treatment. PM4S induced a gradually decreased expression of *FXR* (Figure 7a–c); in addition, PM5S also reduced the relative expression of FXR-targeted genes (*SHP* and *FGF19*). These results suggest that PM4S functions as an FXR antagonist in regulating BA metabolism.

### 2.4. PM5S Function as Antagonist of FXR Signaling Pathway

We further investigate the role of PM5S on hepatocyte BA metabolism. Primary pig hepatocytes were treated with increasing concentrations of PM5S. In contrast with PM4S, PM5S induced a gradually increased expression of FXR targeted genes, including *SHP* and *BSEP* (Figure 8a,b). Since CDCA is the dominant primary BA [19], we further investigated the effect of PM5S on BA metabolism in primary pig hepatocytes co-treated with CDCA. CDCA treatment alone increased the relative expression of FXR targeted genes (*SHP* and *BSEP*) from 50–100 μmol/L (Figure 9), and thus we selected 50μmol/L in the following study. The CDCA treatment resulted in a 5–10-fold increase in BSEP mRNA levels, and a 8-fold decrease in CYP7A1 mRNA levels relative to control cells (Figure 10b,e). However, the relative expression of *BSEP* was decreased with the increasing of PM5S, whereas the expression of *CYP7A1* and TBA levels were increased with the increasing concentration of PM5S (Figure 10a,b,e). The relative expression of *MDR3* was also gradually decreased with the increasing of PM5S (Figure 10f), though the relative expression of *OSTβ* was not significantly changed (Figure 10d). Consistently, the FXR and its target genes *SHP* and *FGF19* levels were also increased after CDCA treatment, whereas it gradually decreased when co-treated with PM5S (Figure 11a–c). These results suggest that PM5S has FXR partial agonist and antagonist activity.

## 3. Discussion

Studies of pregnant women and rodents have revealed the physiological change of BA and reproductive hormones during pregnancy, particularly the role of estrogen or its glucuronidated forms [1,22,26]. However, glucuronidation was a minor pathway for steroid metabolism in humans, rodents, minipigs, and pregnant sows [27,28]. The most important finding in this study was that the dysregulation of BA homeostasis was associated with changes in PMSs metabolism during the latter stages of pregnancy in sows. Consistent with increased serum BA in pregnant women and mice [1,29], TBA levels measured by UPLC-MS/MS profiling and enzymatic assay both increased from G60 to G90. However, unlike the continuously high levels of BA in cholestasis patients before farrowing [30], TBA levels deceased from G90 to L0, though still higher than that at initial time (G60). Several studies demonstrated the role of PMSs on the dysregulation of BA homeostasis during pregnancy [13,21,31]. The positive correlation between the serum PM4S/PM5S and the maternal and fetal TBA were also demonstrated, respectively. Consistent with the results in human primary hepatocytes [13], the partial agonist role of PM5S on FXR-mediated BA homeostasis was also confirmed in pig primary hepatocytes. More importantly, previous studies have demonstrated the inhibition role of PM4S on BA reabsorption in primary human hepatocytes [32] and BA output in rat liver [33] but the detailed mechanism was still unknown; the antagonist role of PM4S on FXR-mediated BA homeostasis was first demonstrated in primary pig hepatocytes. Thus, the decreased levels of PM4S and PM5S from G90 to L0 may well explain the changing trend of TBA from G90 to L0. In addition, as the main downstream of FXR, FGF15 (ortholog of FGF19 in human and pigs), regulating hepatic BA metabolism in mice, was mainly through intestinal FGF15-FGFR4 pathway [34,35], whereas FGF19 was also expressed in human primary hepatocytes and regulated BA synthesis through CYP7A1 [36]. The high expression of *FGF19* in pig hepatocytes in this study indicated that pregnant sows are a promising animal model for the investigation of the pathogenesis of cholestasis during pregnancy.

Though the strong relationships between PMSs and abnormally elevated serum BA have been well established in a series of studies [37,38], limited studies were performed on the source and synthesis pathway of PMSs. A recent study performed on healthy pregnant women showed that the relative levels of PMSs in the human placenta was higher than that in serum of both mother and fetuses [14], which indicated the important role of placenta in PMSs synthesis. Consistent with this study, the higher relative levels of placental *SULT2A1*, responsible for sulphonating hydroxysteroids, such as pregnenolone, at G90 than at G60 and L0, indicated the potential role of the placenta SULT2A1 in PMSs synthesis. In addition to the placenta, gut microbiota was also shown to play an important role in PMSs desulfation. Gut microbiota, especially in the cecum and colon, possess steroid sulfatase (STS) activity, which is responsible for the hydrolysis of aryl and alkyl steroid sulfates, and thus contributes to desulfating steroid sulfates [39,40,41,42]. A recent study in ICP patients also revealed the strong relationship between gut microbiota and ICP [43]. The alleviation role of the probiotic *Lactobacillus rhamnosus* GG in BA accumulation and liver injury induced by PM5S in a mice model [44] further indicated the regulation role of gut microbiota on PMSs metabolisms. Further study is warranted to explore the specific relationship among gut microbiota, PMSs, and BA metabolisms in pregnant sows and women.

## 4. Materials and Methods

### 4.1. Materials

Methanol, acetonitrile, isopropanol, methanoic acid, fetal bovine serum, and DMEM medium were purchased from Thermo Fisher (Fairlawn, NJ, USA). Progesterone, progesterone-d9, allopregnanolone sulfate (PM4S), and epiallopregnanolone sulfate (PM5S) were purchased from Steraloids (Newport, RI, USA). RNAiso Plus reagent, Reverse transcription kit, and SYBR Green kit were purchased from Takara (Dalian, China). The estradiol radioimmunoassay KIT was received from the Beijing North Institute of Biological Technology (Beijing, China). Charcoal, dextran coated, and other biochemical regents were obtained from Sigma (St. Louis, MO, USA), unless otherwise stated. 

### 4.2. Animals and Samples

All experiments on sows were approved by the Animal Care and Use Committee of Animal Nutrition Institute, Sichuan Agricultural University (Ethic Approval Code: SCAUAC201605−1) and were carried out in accordance with the National Research Council’s Guide for the Care and Use of Laboratory Animals, Chinese Order No. 676 of the State Council, date (1 March 2017).

A total of nineteen pregnant sows (nulliparous) with Landrace and Yorkshire background were individually housed in the same room. All sows with about the same body weight (90.0 ± 1.7 kg) were checked daily for estrus with a mature boar from 26 weeks of age, the first detection of standing estrus was taken as the first estrus. Sows were intracervical inseminated by using semen from Duroc boars at the third estrus and were inseminated every 12 h thereafter 3 times. The first day of mating was taken as day 0 of gestation, and the farrowing day was taken as L0. The establishment of pregnancy was further verified through an ultrasound examination at about 25 days post-insemination. The ultrasound examination were made with an all-digital ultrasound diagnostic system (WED-180, Shenzhen Well.D Medical Electronics Co., Ltd., Shenzhen, China), which was equipped with an abdominal convex probe (3.5 MHz), the maximum display depth was 250 mm. The ultrasound examination was performed at the right abdominal wall and near the third caudal mammary gland. The artificial lighting schedule was provided from 08:00 to 18:00 and ambient temperature was kept at 16–25 °C. 

Since the dysregulation of BA homeostasis mainly occurred during the second and third trimester of pregnancy [8], this study was conducted from G60 until L0. A total of nineteen pregnant sows were successfully inseminated. The maternal peripheral blood was continuously sampled after 8-h fasting from seven sows at G60, G75, G90, G105, and L0. Fetal blood and placentas were sampled from six pregnant sows after 8-h fasting at G60 and G90, respectively. When pregnant sows were laparotomized and under deep isoflurane-induced anesthesia. Briefly, atropine sulfate and sumiannin II were injected through the ear vein to complete anesthesia induction. After the sows were effectively anesthetized, they were fixed on the operating table and kept under anesthesia using isoflurane through a respiratory anesthesia mask and the fetal blood of umbilical cord vein and placentas was sampled from fetuses with average body weight. The placental tissue samples, which surrounded the cervix and utero-tubal junction, were immediately flash frozen in liquid nitrogen and then stored at −80 °C. Fetal blood and placental samples were also collected from six sows at the L0 immediately after delivery. The fetal serum samples were acquired after centrifuge at 3000× *g* for 10 min and then stored at −80 °C.

### 4.3. Cell Culture and Treatments

Primary pig hepatocytes were isolated from three female pigs at 2–5 day old using a 2-step perfusion method with a minor modification [45,46]. Briefly, a blood vessel of moderate size was incubated, and the pig liver lobes were perfused with a perfusion buffer for 10 min, the buffer was replaced with collagenase-containing buffer for about 10 min. The liver lobes were placed on a sterile flat plate and 100 mL culture medium with 10% fetal bovine serum was added to terminate the collagenase digestion. Cells were then filtered through a 100-μm mesh following twice washing with a culture medium after removing the liver capsule, blood vessels, fat, and connective tissue. Cell concentration was determined and 1.5 × 10^5^ per well were plated in 24-well plates or 5 × 10^5^ per well were plated in 6-well plates. Cells were grown in complete medium containing phenol red-free DMEM medium, 2.5% dextran-coated, charcoal-stripped fetal bovine serum.

### 4.4. TBA Analysis

The TBA was measured by the same enzymatic cycling method assay kits (Kehua Bio-Engineering Co., Ltd., Shanghai, China) in Model 3100 automatic biochemical analyzer (Hitachi, Tokyo, Japan), the detection range was 1–180 μmol/mL, and the sensitivity was below 5 pg/mL, the intra-assay variable coefficient was lower than 5%, the inter-assay variable coefficient was lower than 10%. 

### 4.5. Estradiol, Progesterone and Sulfated Progesterone Metabolites

Estradiol was measured by the same estradiol radioimmunoassay KIT according to the manufacturer’s specifications (Iodine [125I] Estradiol radioimmunoassay Kit, Beijing North institute of Biotechnology, Beijing, China), the detection range was 6–1000 pg/mL, and the sensitivity was below 5 pg/mL, the intra-assay variable coefficient was lower than 10%. An ultra-performance liquid chromatography coupled to tandem mass spectrometry (UPLC-MS/MS) system (ACQUITY UPLC-Xevo TQ-S, Waters Corp., Milford, MA, USA) was used to quantitate progesterone, PM4-S, and PM5-S adapted from the previous literature [14]. The following multiple reaction monitoring (MRM) transitions were monitored in this study for the analytes: progesterone (315.2/97.0), PM4-S (397.1/97.2), and PM5-S (397.1/97.2), respectively. These steroids were separated on an ACQUITY UPLC BEH C18 1.7 µM analytical column (2.1 mm × 100 mm) at 40 °C using the following gradient: hold at 50% B for initial 3 min, increase to 80% for 9 min, and raise to 100% for 0.5 min prior to switching to the initial condition. The mobile phase B was methanol with 5 mM ammonium acetate and the mobile phase A was water with 5 ammonium acetate (pH = 3.0 adjusted with acetic acid). The flow rate was 0.4 mL/min and the injection volume was 5 µL.

### 4.6. Real-Time RT-PCR

All primers were designed through primer BLAST from NCBI or used as previously described [28], the primers are shown in Appendix A. The total RNA of frozen samples was extracted using the RNAiso Plus reagent, and the complementary DNA (cDNA) was synthesized using a reverse transcription kit. Real-time quantitative PCR (RT-qPCR) was performed on a CFX96 Real-Time PCR Detection System (Bio-Rad, Hercules, CA, USA) to quantify mRNA expression with a commercial SYBR Green kit. β-actin was used as the internal control. Relative expression levels of the target genes were calculated using the 2^−ΔΔCT^ method [47].

## 5. Data Analysis

Data are expressed as means ± SE. The relative expression mRNA levels between CON and CDCA were analyzed by Student’s *t*-test of GraphPad Prism (8.0.2 version). Data passing the normality test were analyzed by a one-way ANOVA of GraphPad Prism and means were compared using the Student–Newman–Keuls test. Correlation analysis was carried out using the CORR procedure of the SAS statistical package (V9.4, SAS Institute Inc., Cary, NC, USA) to study the relationship between PMSs and BA levels in both maternal and fetal serum at three timepoints (G60, G90 and L0) together. *p* < 0.05 was considered statistically significant.

## 6. Conclusions

Taken together, the findings of this study demonstrate that sulfated progesterone metabolites PM4S and PM5S are highly correlated with BA homeostasis in both mothers and fetuses. PM4S and PM5S regulate BA homeostasis through their functions as FXR inhibitor and antagonist, respectively. Together with the high relative expression of *FGF19* in pig primary hepatocytes, pregnant sows are a promising animal model to investigate the pathogenesis of cholestasis during pregnancy.

## Figures and Tables

**Figure 1 ijms-23-06496-f001:**
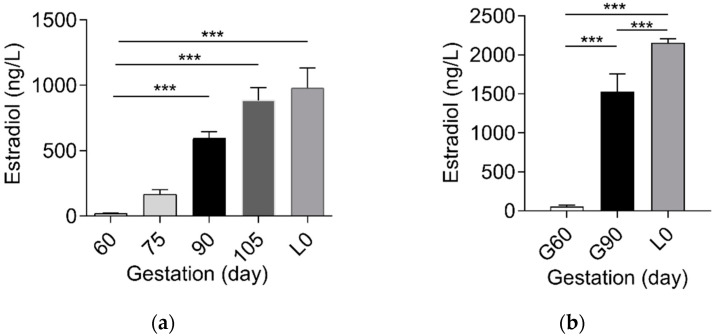
(**a**) Comparison of maternal serum estradiol at G60, day 75 of gestation (G75), G90, day 105 of gestation (G105), and L0, (**b**) and fetal serum estradiol at G60, G90, and L0 (n = 6–7/group). Data are shown as means ± SE, *** *p* < 0.001.

**Figure 2 ijms-23-06496-f002:**
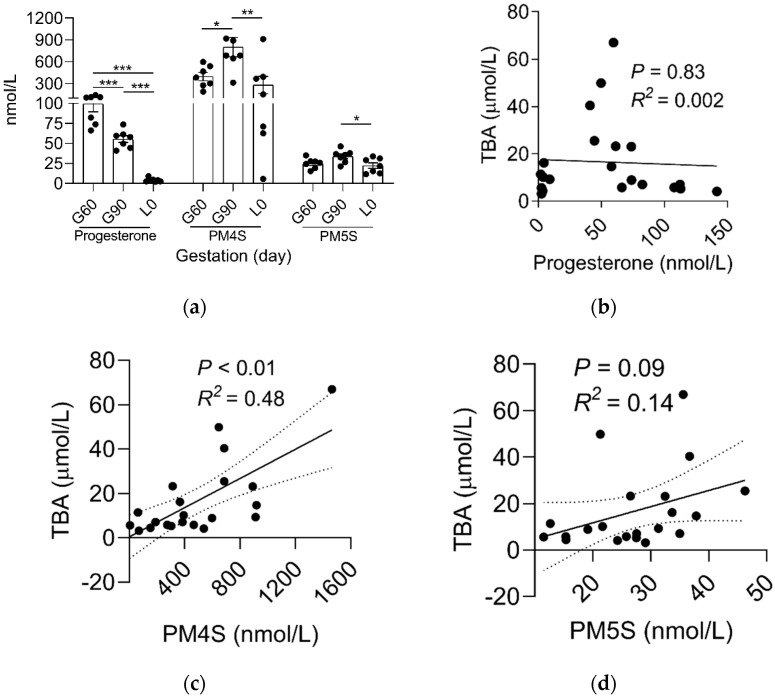
The comparison of maternal serum (**a**) progesterone, PM4S, and PM5S at G60, G90, and L0. n = 7/group. The linear correlation of (**b**) progesterone, (**c**) PM4S, and (**d**) PM5S and maternal serum TBA. n = 21. Data are shown as means ± SE, * *p* < 0.05, ** *p* < 0.01, *** *p* < 0.001.

**Figure 3 ijms-23-06496-f003:**
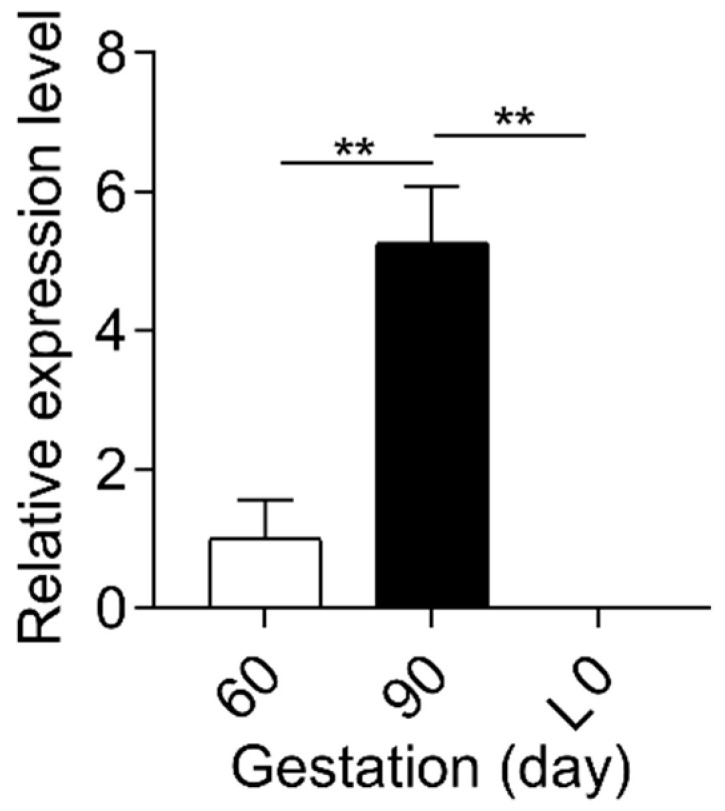
Relative expression levels of *SULT2A1* in placentas at G60, G90, and L0, respectively. n = 3/group. Data are shown as means ± SE, ** *p* < 0.01.

**Figure 4 ijms-23-06496-f004:**
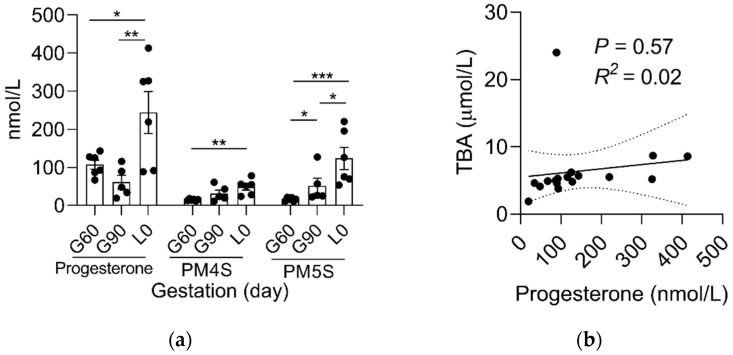
Comparison of fetal serum (**a**) progesterone, PM4S, and PM5S at G60, G90, and L0 (n = 5–6/group). The linear correlation of (**b**) progesterone, (**c**) PM4S, and (**d**) PM5S and fetal serum TBA (n = 17). Data are shown as means ± SE, * *p* < 0.05, ** *p* < 0.01, *** *p* < 0.001.

**Figure 5 ijms-23-06496-f005:**
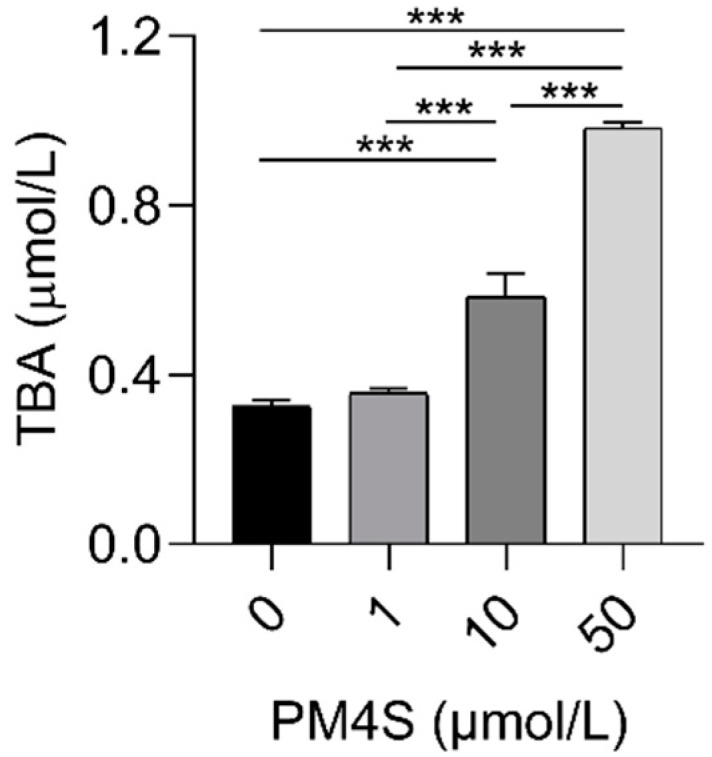
The TBA levels of pig primary hepatocytes culture medium with increasing dose of PM4S (0, 1, 10, and 50 μmol/L), respectively. n = 3/group. Data are shown as means ± SE, *** *p* < 0.001.

**Figure 6 ijms-23-06496-f006:**
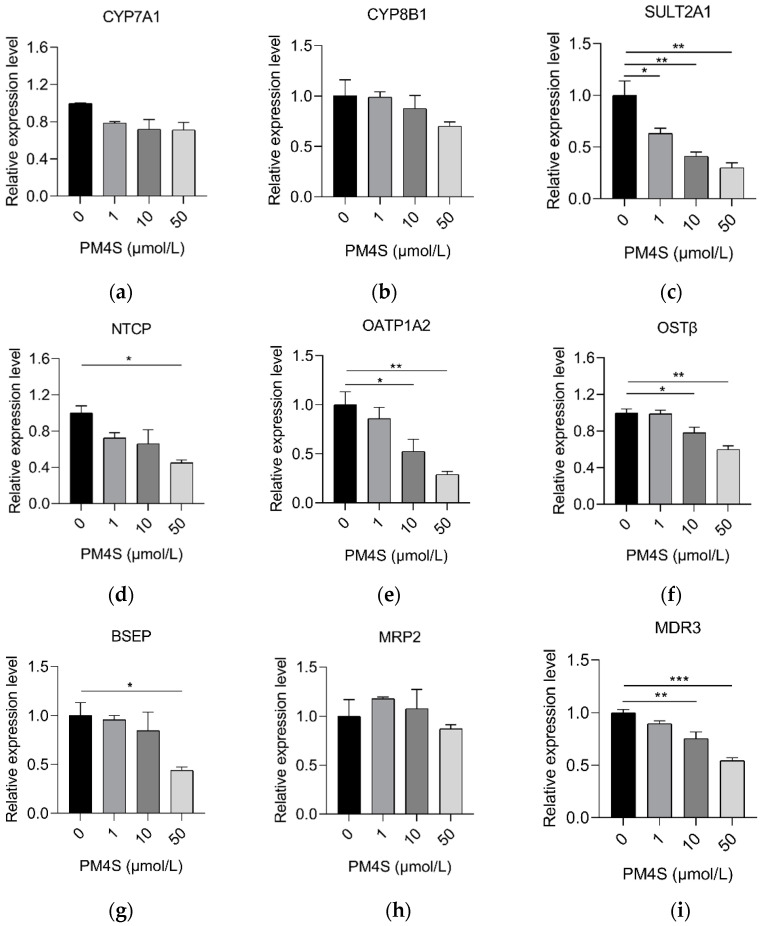
Relative expression mRNAs of pig primary hepatocytes involved in (**a**,**b**) BA synthesis, (**c**) BA sulfation, (**d**,**e**) BA reabsorption, (**f**) BA efflux, (**g**,**h**) BA biliary secretion, and (**i**) phosphatidylcholine secretion, respectively. n = 3/group. Data are shown as means ± SE, * *p* < 0.05, ** *p* < 0.01, *** *p* < 0.001.

**Figure 7 ijms-23-06496-f007:**
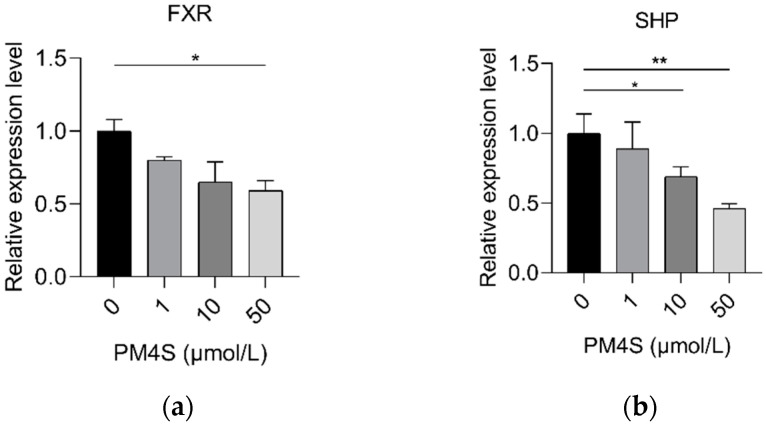
Relative expression of (**a**) *FXR*, (**b**) *SHP*, and (**c**) *FGF19* in pig primary hepatocytes with an increasing dose of PM4S, respectively. n = 3/group. Data are shown as means ± SE, * *p* < 0.05, ** *p* < 0.01, *** *p* < 0.001.

**Figure 8 ijms-23-06496-f008:**
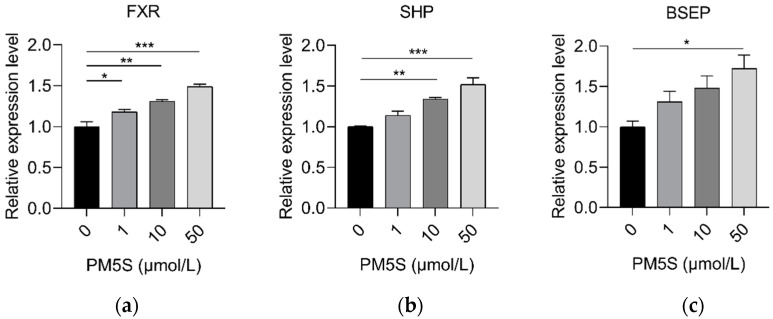
Relative expression of (**a**) *FXR*, (**b**) *SHP*, and (**c**) *BSEP* in pig primary hepatocytes with increasing doses of PM4S (0, 1, 10, and 50 μmol/L), respectively. n = 3/group. Data are shown as means ± SE, * *p* < 0.05, ** *p* < 0.01, *** *p* < 0.001.

**Figure 9 ijms-23-06496-f009:**
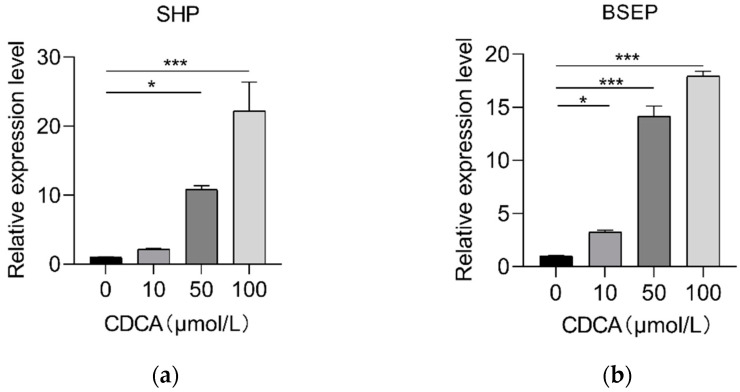
Relative expression of (**a**) *SHP* and (**b**) *BSEP* in pig primary hepatocytes with increasing doses of CDCA (0, 10, 50, and 100 μmol/L), respectively. N = 3/group. Data are shown as means ± SE, * *p* < 0.05, *** *p* < 0.001.

**Figure 10 ijms-23-06496-f010:**
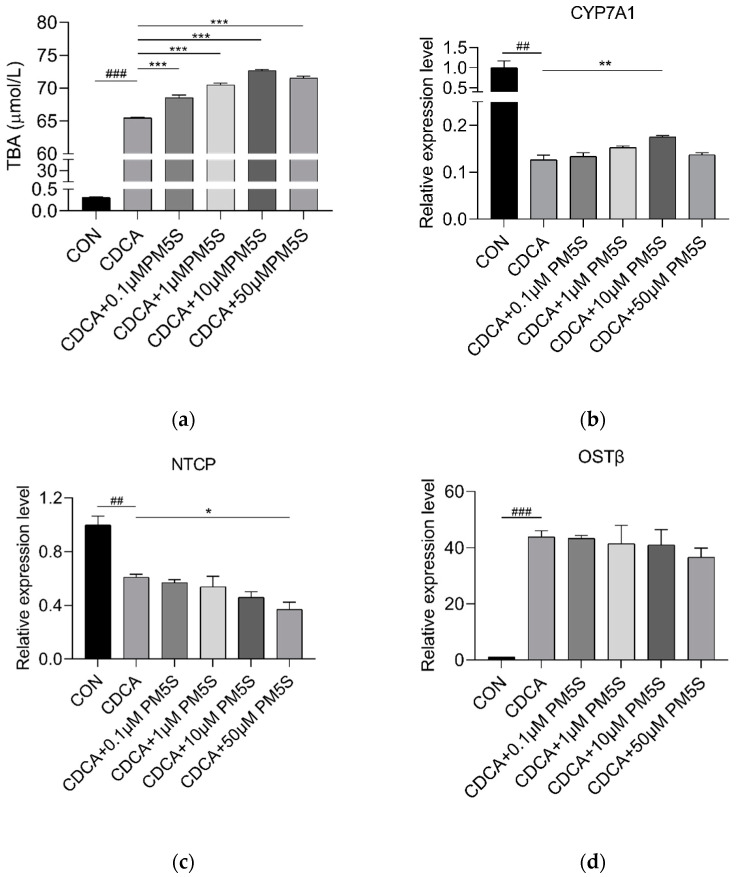
Relative expression of (**a**) cell medium TBA levels, (**b**) *CYP7A1*, (**c**) *NTCP*, (**d**) *OSTI* (**e**) *BSEP* and (**f**) *MDR3* in pig primary hepatocytes co-treated with CDCA (0, 50 μmol/L) and PM5S (0, 0.1, 1, 10, 50 μmol/L). n = 3/group. Data are shown as means ± SE, ^##^
*p* < 0.01 for CON versus CDCA group, ^###^
*p* < 0.001 for CON versus CDCA group. * *p* < 0.05 for CDCA versus CDCA and PM5S co-treatment group, ** *p* < 0.01 for CDCA versus CDCA and PM5S co-treatment group, *** *p* < 0.001 for CDCA versus CDCA and PM5S co-treatment group.

**Figure 11 ijms-23-06496-f011:**
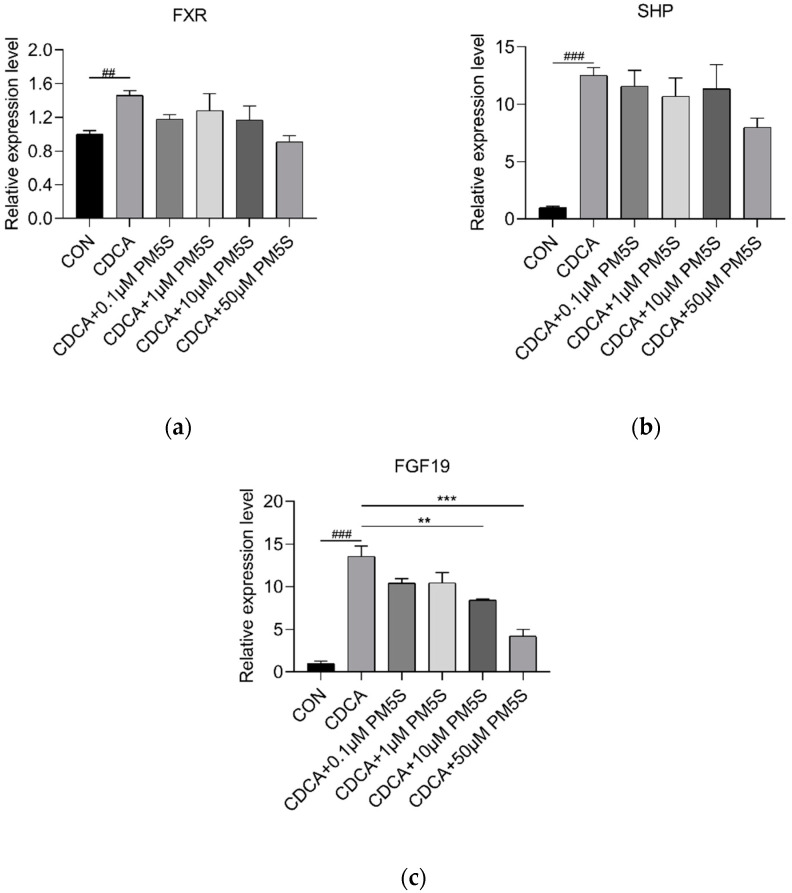
Relative expression of (**a**) *FXR*, (**b**) *SHP* and (**c**) *FGF19* in pig primary hepatocytes co-treated with CDCA (0, 50 μmol/L) and PM5S (0, 0.1, 1, 10, 50 μmol/L). n = 3/group. Data are shown as means ± SE, ^##^
*p* < 0.01 for CON versus CDCA group, ^###^
*p* < 0.001 for CON versus CDCA group. ** *p* < 0.01 for CDCA versus CDCA and PM5S co-treatment group, *** *p* < 0.001 for CDCA versus CDCA and PM5S co-treatment group.

## Data Availability

All data generated in the study are included in the published article and its additional files. The datasets generated in the current study are available from the corresponding author on reasonable request.

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
