# Peer review of "Maternal and Fetal Bile Acid Homeostasis Regulated by Sulfated Progesterone Metabolites through FXR Signaling Pathway in a Pregnant Sow Model"

_ijms, 2022, doi:10.3390/ijms23126496_

Round 1

Reviewer 1 Report

Interesting article with data from pregnant sows and using different methods to answer the scientific questions they propose.

As a doubt, I wonder how much of the PSM will be generated via enzymatic and other routes. I don't know if there are any studies on this.

Maybe figure 2 should be fig 3. Fig 1c&d can be in supplementary data. The results of quadratic correlations seem to come from linear correlations (fig 1, 3). Please, review this and add this information in the M&M section. If it is not explaining, you have all the time points together in the correlations; why not separated? or at least comment on it.
It would be interesting to see the levels of TBA in the three time points. There are no data about the profile of BA; please, at least, add as supplementary data.
Correlation with individual BA would be interesting to know.

The discussion of the results is direct but maybe too simple. It may include information from studies about this topic and take account of the factor of the microbiota that is quite important for bile acids and sexual hormones.

https://www.frontiersin.org/articles/10.3389/fcimb.2021.635680/full
https://www.sciencedirect.com/science/article/pii/S0006291X1931839X?via%3Dihub

The breed or crossbreed of sows is unknown. Sample collection methods are not well described. The methods for correlations have to be indicated in the M&M section. In general, this section could have more details in several subsections.
I think a method to reduce error type 1 is needed

Please, change the format of the original images (scn) to another more common.

Review Acknowledgments, spaces, points,...

Author Response

Response to Reviewer 1 Comments

Point 1:  Interesting article with data from pregnant sows and using different methods to answer the scientific questions they propose.

As a doubt, I wonder how much of the PSM will be generated via enzymatic and other routes. I don't know if there are any studies on this.

Response 1: Thanks for your valuable comments. The elevated serum levels of PMSs in patients with intrahepatic cholestasis of pregnancy (ICP) have been well studied (LING et al.,Hepatology 1997; LING et al.,Journal of Hepatology 1997 ). However, limited studies were performed on the source and synthesis pathway of PMSs. A recent study perfomed on ICP patients showed that the relative levels of PMSs in human placenta was higher than that in serum of both mother and fetuses, which imply the important role of placenta in PMSs synthesis (Maria et al.,Br J Clin Pharmacol 2014). Consistent with above study, the higher relative mRNA levels of placental SULT2A1, responsible for sulphonating hydroxysteroids such as pregnenolone, at G90 compared with G60 and L0 in our study, indicating the potential role of placental SULT2A1 in PMSs synthesis. Moreover, the different metabolism pattern in mother and fetuses during pregnancy would also make some contribution to this field.

Point 2: Maybe figure 2 should be fig 3. Fig 1c&d can be in supplementary data. The results of quadratic correlations seem to come from linear correlations (fig 1, 3). Please, review this and add this information in the M&M section. If it is not explaining, you have all the time points together in the correlations; why not separated? or at least comment on it.

Response 2: Thanks for your valuable comments. we have changed the position of figure 2 to figure 3 and the transferred Figure 1 c& d to the supplementary figure S1 according to your suggestion.

The description of quadratic correlations (Fig 1,3) was a mistake, and we have changed it to linear corrleations according to your suggestion. Thanks again for your kind help.

Consistent with result in pregnant women and mice, our previous study showed maternal BA metabolism in pregnant swine was dysregulated with the advance of pregnancy (Peng Wang et al, Am J Physiol Gastrointest Liver Physiol 2019), in order to clarify the relationship between TBA and PMSs during pregnancy, we further analysized the correlation of PMSs and TBA levels at all the time point.

Point 3: It would be interesting to see the levels of TBA in the three time points. There are no data about the profile of BA; please, at least, add as supplementary data.

Correlation with individual BA would be interesting to know.

Response 3: Thanks for your valuable comments. We have added the TBA levels in the supplementary data(Supplementary Figure S2), unfortunatley, we failed to measure the BA profile in this study, we will try to clarify the relationship between PMSs and BA profile in our further studies, thanks again for your kind help and suggestion.

Point 4: The discussion of the results is direct but maybe too simple. It may include information from studies about this topic and take account of the factor of the microbiota that is quite important for bile acids and sexual hormones.

https://www.frontiersin.org/articles/10.3389/fcimb.2021.635680/full

https://www.sciencedirect.com/science/article/pii/S0006291X1931839X?via%3Dihub

Response 4: Thanks for your valuable comments. we have modified this part and take account of the microbiota in this field according to your suggestion. Part of discussion was shown as follows:

Though the strong relationship between PMSs and abnormally elevated serum BA have been well established in series studies[34,35], limited studies were performed on the source and synthesis pathway of PMSs. A recent study performed on healthy pregnant women showed that the relative levels of PMSs in human placenta was higher than that in serum of both mother and fetuses[36], which imply the important role of placenta in PMSs synthesis. Consistent with this study, the higher relative levels of placental SULT2A1, responsible for sulphonating hydroxysteroids such as pregnenolone, at G90 compared with G60 and L0, indicating the potential role of placenta SULT2A1 in PMSs synthesis. In addition to placenta, gut microbiota was also played an important role in PMSs desulfation. Gut microbiota, especially in the cecum and colon, possess steroid sulfatase (STS) activity, which is responsible for hydrolysis of aryl and alkyl steroid sulfates, and thus contribute to desulfating steroid sulfates[37-40]. Recent study in ICP patients also revealed the strong relationship between gut microbiota and ICP[41]. The alleviation role of Probiotic Lactobacillus rhamnosus GG in BA accumulation and liver injury induced by PM5S in a mice model[42] further verify the regulation role of gut microbiota on PMSs metabolism. Further study is warranted to explore the specific re-lationship among gut microbiota, PMSs and BA metabolism in pregnant sows and women.

Point 4: The breed or crossbreed of sows is unknown. Sample collection methods are not well described. The methods for correlations have to be indicated in the M&M section. In general, this section could have more details in several subsections.

I think a method to reduce error type 1 is needed

Please, change the format of the original images (scn) to another more common.

Review Acknowledgments, spaces, points,...

Response 4: Thanks for your valuable comments. we have reviewed carfefully about the point you have raised, and added the breed infromation, sampling collection method in the M&M section. We also add the disscusstion and acknowledgements part.

A total of nineteen pregnant sows (nulliparous) with Landrace and Yorkshire background were individually housed in the same room. All sows with about 90 kg were checked daily for estrus with a mature boar from 26 weeks of age, the first detection of standing estrus was taken as the first estrus. Sows were intracervical inseminated by using semen from Duroc boars at the third estrus, and were inseminated every 12 h thereafter for 3 times. The first day of mating was taken as day 0 of gestation, and the day of farrowing day was taken as L0. The establish of pregnancy was further verified through ultrasound examination at about 25 days post-insemination. The artificial lighting schedule was provided from 0800 to 1800h and ambient temperature was kept at 16-25°C.

Since dysregulation of BA homeostasis was mainly occurred during the second and third trimester of pregnancy [8], this study was conducted from G60 until L0. A total of nineteen pregnant sows were successfully inseminated. The maternal peripheral blood was continuously sampled after 8-hours fasting from seven sows at G60, G90 and L0. Fetal blood and placentas were sampled from six pregnant sows after 8-hours fasting at G60, and G90, respectively. when pregnant sows were laparotomized and under deep isoflurane-induced anesthesia. Briefly, atropine sulfate and sumiannin II were injected through ear vein to complete anesthesia induction. After the sows were effectively anesthetized, they were fixed on the operating table and maintained anesthetized with isoflurane through a respiratory anesthesia mask, the fetal blood of umbilical cord vein and placentas were sampled from fetuses with average body weight. The placental tissue samples, which surrounded the cervix and utero-tubal junction, were immediately flash frozen in liquid nitrogen and then stored at -80℃. Fetal blood and placental samples were also collected from six sows at the L0 immediately after delivery. The fetal serum samples were acquired after centrifuge at 3000 g for 10 min and then stored at -80℃.

Reviewer 2 Report

The presented manuscript is interesting, however, it is confusing. Some of the issues that require clarification are listed below in the detailed comments. Please clarify them before I go to the results review and discussion, thus based on the work presented in this way, I am not able to judge whether your research has been carried out correctly.

The abstract section should be rewritten.
L 21 "reproductive hormone"  all reproductive hormones or selected?
L 21 "mother and fetuses" circulation? peripheral blood? Please, specify.
L 22 "to reveal its potential mechanisms" mechanisms of action? On what? What kind of pathway did you investigate?
L 22 "UPLC-MS/MS" all abbreviations have been expended for the first time in the text.
L 22 "we first" use passive voice. Please, correct it throughout the whole manuscript body. (L 24 "We also identified"; L 26 "We confirmed"; L30 "Our results"...; L 270 "we previously found")
L 23 "PM4S and PM5S" all abbreviations have been expended for the first time in the text.
L 24 "compartments of pregnant swine" please, clarify what kind of tissues you investigated? Moreover, the number of studied animals has been added. More methodological details are required here, including the details of correlations calculation.
L 26 "FXR" all abbreviations have been expended for the first time in the text.
L 26-28 How it was confirmed?
L 28 "FGF19" all abbreviations have been expended for the first time in the text.
L 29 "pig hepatocytes" There is no data above, that the liver tissue was investigated. Please, add more methodological details in the abstract section, as it is not easy to follow what and how you investigated.
L 30-32 I can not agree as long as it is not clear how you came to these conclusions.

Introduction section
Please, check in the guidelines for the authors if all references should appear in the round or square parenthesis.
L 37 "mother" mother's tissue? mother's organism? please, be more precise. 
L 38 "BA" Besides the abstract section, in the manuscript body all abbreviations have been expended for the first time in the text.
L 42 "fetal compartments" please, clarify what kind of tissue you mean.
L 46 "interventions to alleviate maternal elevated BA may improve fetal outcomes" please, be more precise. What kind of interventions? maternal elevation of BA where? in circulation? peripheral blood? selected tissue?
L 47 "increased BA" where?
L 48 "Steroid hormone" Did you mean one specific steroid hormone or steroid hormones generally or reproductive steroid hormones? If only estrogens and progesterone are the subject, this should be explained here.
L 50 "estrogen" there is no one estrogen in the circulation and tissues, please, specify.
L 50 Please, explain the involvement of estrogens in the regulation of BA metabolism. Which regulatory pathway has been identified so far?
L 51 Did progesterone involve in regulating BA metabolism through specific receptors? Please, introduce progesterone as well as estrogens concerning the regulatory pathway before introducing its metabolites.
L 51 "women and animal" only female rats or also other animals?
L 51-52 "a role for sulfated progesterone metabolites (PMSs) in dysregulation of maternal BA metabolism in ICP patients" describe briefly this role.
L 54 "estrogen" there is no one estrogen in the circulation and tissues, please, specify.
L 58 "estrogen" estrogens?
L 58 "These factors" Which factors? similarities?
L 59 "that results observed in human hypercholanemia have important implications for pregnant swine and fetal survival" I can not agree. The base on the similarities in the recent research you can not conclude any implications. Moreover, please decide did you use the porcine specimen as a model for humans or did you follow human-specific pathways in pigs to use them in the pig breeding?
L 61 "rodents" or female mice?
L 61 "swine" or sow? as it specifically refers to the female pig. Please, use it throughout the whole manuscript body.
L 67 "mother and fetuses" tissues? peripheral blood? Please, specify. 
L 68 "regulation mechanism of PMSs on BA metabolism" as the investigation of the regulation mechanism is very complex, I would be careful with formulating such a goal and focus on the actual goal of the work, which is to compare the concentration of test compounds in the blood/tissue of sows and fetuses at 60, 90 and 115 (?) days of pregnancy.

Material and method section
L 267 "All pregnant swine" how many sows were used in the experiment? Provide n. Please, also provide more details about used sows - age, body mass, breed.
L 268 "artificially inseminated" when, concerning the estrus or ovarian cycle, were sows inseminated?
L 271-272 "from day 60 of gestation (G60)" How and when the pregnancy was confirmed? How many AI sows were pregnant confirmed? Was it possible to get pregnant each time after the first AI?
L 272-277 It is not clear how many sows were included in each group. Do I understand correctly that you only took samples three times? On days 60, 90, and 114-116(115) of pregnancy? How long was the pregnancy of each of the pigs tested? Which day of pregnancy was day L0? Was each sampling associated with the termination of pregnancy?
Do I understand correctly that on day G60 samples were taken from 6 sows, on day G90 also from 6 sows, and day L0 from 9 sows?
Explain in detail what exactly (what tissue, blood) was taken from each sow and how was it stored?
Explain in detail what exactly (what tissue, blood) was taken from each fetus and how was it stored? 
How many fetuses were present in the uterus of each sow at the time of sampling. How many fetuses were sampled? From all, a dozen, a few, or one randomly assigned?
L 273 "and the farrowing" or? were the same sows used twice or three times?
L 275 Did 8-hours fasting not affect your BA metabolism? Were the sows also subjected to 8-hours of fasting at the time of L0 sampling? If not, can you justify the desirability of comparing the results obtained in this way?
L 276 Detail of the laparotomy protocol including deep isoflurane-induced anesthesia should be provided. 
L 279 "2-5 day old female pigs" How many pigs were used?
L 291 Please, provide the manufacture details, numbers of kits, the sensitivity, intraassay CV, and interassay CV.
L 297-299 Please, provide the manufacture details, numbers of estradiol kits, the sensitivity, intraassay CV, and interassay CV. Did you investigate 17ß-estradiol?
L 315 and 318 Please, provide the kit details.
L 318 Please, provide the investigated gene expression list including used forward and reverse primer sequences.
L 319 In which samples/group the target gene expression was considered as ΔCt Control Value?
L 334 Excepting PROC GLM, please provide the test used for univariate distributions testing. In the compared data series, has always at least one data series shown a non–Gaussian distribution? Please, describe in detail what you compared with what and how, because based on the presented description I am not able to judge whether the statistical analysis was carried out correctly.
Both in the abstract and for the aim of the study sections, you indicated that the correlation between the data series was examined, while in the place appropriate for this description there is no data about carrying out the relevant calculations.

Author Response

Response to Reviewer 2 Comments

Point 1: The presented manuscript is interesting, however, it is confusing. Some of the issues that require clarification are listed below in the detailed comments. Please clarify them before I go to the results review and discussion, thus based on the work presented in this way, I am not able to judge whether your research has been carried out correctly.

The abstract section should be rewritten.

L 21 "reproductive hormone" all reproductive hormones or selected?

Response 1: Thanks for your valuable comments, we have changed the “reproductive hormone” to “investigate the intercorrelation of reproductive hormone, including estradiol, progesterone and sulfated progesterone metabolites (PMSs)” in the abstract section.

Point 2:

L 21 "mother and fetuses" circulation? peripheral blood? Please, specify.

Response 2: Thanks for your valuable comments, we have changed the words "mother and fetuses" to “in the peripheral blood of mother and fetuses”

Point 3:

L 22 "to reveal its potential mechanisms" mechanisms of action? On what? What kind of pathway did you investigate?

Response 3: Thanks for your valuable comments, we have deleted these words and modified the sentense in Line 78-81 “the aim of the current study was to reveal the relationship between PMSs and BA in peripheral blood of mother and fetuses at G60, G90 and L0, and the regulation mechanism of PMSs on BA metabolism through FXR signaling pathway.”

Point 4:

L 22 “UPLC-MS/MS” all abbreviations have been expended for the first time in the text.

Response 4: Thanks for your valuable comments, we have changed the “UPLC-MS/MS” to “ultraperformance liquid chromatography-tandem mass spectrometry (UPLC-MS/MS)” in the abstract section.

Point 5:

L 22 "we first" use passive voice. Please, correct it throughout the whole manuscript body. (L 24 "We also identified"; L 26 "We confirmed"; L30 "Our results"...; L 270 "we previously found")

Response 5: Thanks for your valuable comments, we have carefully corrected the grammar you have raised.

    For example, in line 22-28, we have changed the word from “By using newly developed UPLC-MS/MS methods, we first confirmed the presence of two of sulfated progesterone metabolites, PM4S and PM5S, in maternal and fetal compartments of pregnant swine. We also identified that pregnancy-associated maternal BA homeostasis was correlated with maternal serum PM4S levels, whereas fetal BA homeostasis was correlated with fetal serum PM5S levels. We confirmed the partial agonist role of PM5S on FXR-mediated BA homeostasis in pig primary hepatocyte model, and first revealed the antagonist role of PM4S on FXR-mediated BA homeostasis” to “Allopregnanolone sulphate (PM4S) and epiallopregnanolone sulphate (PM5S) were firstly identified in maternal and fetal peripheral blood of pregnant sows, by using newly developed ultraperformance liquid chromatography-tandem mass spectrometry (UPLC-MS/MS) methods. Correlation analysis showed that pregnancy-associated maternal BA homeostasis was correlated with maternal serum PM4S levels, whereas fetal BA homeostasis was correlated with fetal serum PM5S levels. The antagonist activity role of PM5S on farnesoid X receptor (FXR) mediated BA homeostasis and fibroblast growth factor 19 (FGF19) were confirmed in PM5S and FXR activator co-treated pig primary hepatocytes model, and the antagonist role of PM4S on FXR-mediated BA homeostasis and FGF19 were also identified in PM4S treated pig primary hepatocytes model”.

Point 6:

L 23 "PM4S and PM5S" all abbreviations have been expended for the first time in the text.

Response 6: Thanks for your valuable comments, we have changed the "PM4S and PM5S" to “Allopregnanolone sulphate (PM4S) and epiallopregnanolone sulphate (PM5S)” in the abstract.

Point 7:

L 24 "compartments of pregnant swine" please, clarify what kind of tissues you investigated? Moreover, the number of studied animals has been added. More methodological details are required here, including the details of correlations calculation.

Response 7: Thanks for your valuable comments, we have added the description” A total of nineteen pregnant sows were randomly assigned to day 60, 90 of gestation (G60, G90), and the farrowing day (L0), to investigate the intercorrelation of reproductive hormone, including estradiol, progesterone and sulfated progesterone metabolites (PMSs), and BA in the peripheral blood of mother and fetuses during pregnancy” in the abstract section.

Data analysis was also added in the abstract section. All data were analyzed by PROC GLM, the Shapiro-Wilk was used to test the univariate distributions. The least significant difference test was used to compare the group means when the F test in the analysis of variance table was significant. Correlation analysis was carried out using the CORR procedure of SAS to study the relationship between PMSs and BA levels in both maternal and fetal serum at G60, G90 and L0.

Point 8:

L 26 "FXR" all abbreviations have been expended for the first time in the text.

Response 8: Thanks for your valuable comments, we have changed the “FXR” to “farnesoid X receptor (FXR)” in the abstract.

Point 9:

L 26-28 How it was confirmed?

Response 9: Thanks for your valuable comments, we have added the methods used in this part.

The partial agonist role of PM5S on farnesoid X receptor (FXR) mediated BA homeostasis and fibroblast growth factor 19 (FGF19) were confirmed in PM5S alone, PM5S and FXR activator co-treated pig primary hepatocytes, and the antagonist role of PM4S on FXR-mediated BA ho-meostasis and FGF19 were also identified in PM4S treated pig primary hepatocytes

Point 10:

L 28 "FGF19" all abbreviations have been expended for the first time in the text.

Response 10: Thanks for your valuable comments, we have changed the “FGF19” to “fibroblast growth factor 19 (FGF19)” in the abstract section.

Point 11:

L 29 "pig hepatocytes" There is no data above, that the liver tissue was investigated. Please, add more methodological details in the abstract section, as it is not easy to follow what and how you investigated.

Response 11: Thanks for your valuable comments, the experiment was not performed on the liver tissue, but on pig primary hepatocytes. We have added the methodological details of pig primary hepatocytes in the abstract section.

Part of the modified abstract was shown as follow: The partial agonist role of PM5S on farnesoid X receptor (FXR) mediated BA homeostasis and fibroblast growth factor 19 (FGF19) were confirmed in PM5S alone, PM5S and FXR activator co-treated pig primary hepatocytes, and the antagonist role of PM4S on FXR-mediated BA ho-meostasis and FGF19 were also identified in PM4S treated pig primary hepatocytes.

Point 12:

L 30-32 I can not agree as long as it is not clear how you came to these conclusions.

Response 12: Thanks for your valuable comments, which help improve our manuscripts. We have deleted the conclusion to make it more accurate.

Point 13:

Introduction section

Please, check in the guidelines for the authors if all references should appear in the round or square parenthesis.

L 37 "mother" mother's tissue? mother's organism? please, be more precise.

Response 13: Thanks for your valuable comments, we have modified the reference cited from round to squre parenthesis in the manuscript, we also changed the “mother” to “mother’s organ system”

Point 13:

L 38 "BA" Besides the abstract section, in the manuscript body all abbreviations have been expended for the first time in the text.

Response 13: Thanks for your valuable comments, we have modified this mistake, and checke it throughout the manuscripts.

Point 14:

L 42 "fetal compartments" please, clarify what kind of tissue you mean.

Response 14: Thanks for your valuable comments, we have change the "fetal compartments" to “BA in fetal blood circulation”.

Point 15:

L 46 "interventions to alleviate maternal elevated BA may improve fetal outcomes" please, be more precise. What kind of interventions? maternal elevation of BA where? in circulation? peripheral blood? selected tissue?

Response 15: Thanks for your valuable comments, to make it more precise, we have changed the sentense to “Evidence that disruption of BA homeostasis causes of adverse fetal outcomes suggests that alleviating maternal elevated serum BA may improve fetal outcomes”

Point 16:

L 47 "increased BA" where?

Response 16: Thanks for your valuable comments, we have changed the “increased BA”to “increased maternal peripheral serum BA”.

Point 17:

L 48 "Steroid hormone" Did you mean one specific steroid hormone or steroid hormones generally or reproductive steroid hormones? If only estrogens and progesterone are the subject, this should be explained here.

L 50 "estrogen" there is no one estrogen in the circulation and tissues, please, specify.

L 50 Please, explain the involvement of estrogens in the regulation of BA metabolism. Which regulatory pathway has been identified so far?

Response 17: Thanks for your valuable comments, we have modified the sentense from “Previous studies have demonstrated estrogen involved in regulating BA metabolism through estrogen receptor, such as estrogen receptor α(1, 12)” to “Estrogens and progesterone likely played important role in the pathogenesis of dis-rupted BA homeostasis. Previous studies have demonstrated estrogen, such as estra-diol,estradiol-17β-glucuronide, involved in regulating BA metabolism through estrogen receptor, such as estrogen receptor α”

Point 18:

L 51 Did progesterone involve in regulating BA metabolism through specific receptors? Please, introduce progesterone as well as estrogens concerning the regulatory pathway before introducing its metabolites.

Response 18: Thanks for your valuable comments, there was no report about the relationship between progesterone, except for sulfated progesterone metabolites, and BA metabolism. We have added the sentense: In contrast to estradiol, there was no report about the direct interaction of progesterone and BA metabolism.

Point 19:

L 51 "women and animal" only female rats or also other animals?

Response 19: Thanks for your valuable comments, we have modified “women and animal” to “pregnant women and female mice”.

Point 20:

L 51-52 "a role for sulfated progesterone metabolites (PMSs) in dysregulation of maternal BA metabolism in ICP patients" describe briefly this role.

Response 20: Thanks for your valuable comments, we have modified the sentense to”Recently, studies in pregnant women and female mice have identified epiallopreg-nanolone sulphate (PM5S) reduced BA secretion through inhibiting farnesoid X receptor (FXR) signaling”.

Point 21:

L 54 "estrogen" there is no one estrogen in the circulation and tissues, please, specify.

L 58 "estrogen" estrogens?

Response 21: Thanks for your valuable comments, we have modified the “estrogen” to “estradiol” in both line 54 and 58.

Point 22:

L 58 "These factors" Which factors? similarities?

L 59 "that results observed in human hypercholanemia have important implications for pregnant swine and fetal survival" I can not agree. The base on the similarities in the recent research you can not conclude any implications. Moreover, please decide did you use the porcine specimen as a model for humans or did you follow human-specific pathways in pigs to use them in the pig breeding?

Response 22: Thanks for your valuable comments, we decided to delete these sentense.

Point 23:

L 61 "rodents" or female mice?

L 61 "swine" or sow? as it specifically refers to the female pig. Please, use it throughout the whole manuscript body.

Response 23: Thanks for your valuable comments, we have change the “rodents” to “pregnant mice” and “pregnant swine” to “pregnant sows”.

Point 24:

L 67 "mother and fetuses" tissues? peripheral blood? Please, specify.

Response 24: Thanks for your valuable comments, we have change the “mother and fetuses”to “in peripheral blood of mother and fetuses”.

Point 25:

L 68 "regulation mechanism of PMSs on BA metabolism" as the investigation of the regulation mechanism is very complex, I would be careful with formulating such a goal and focus on the actual goal of the work, which is to compare the concentration of test compounds in the blood/tissue of sows and fetuses at 60, 90 and 115 (?) days of pregnancy.

Response 25: Thanks for your valuable comments, we have carefully modified the "regulation mechanism of PMSs on BA metabolism" to “the regulation mechanism of PMSs on BA metabolism through FXR signaling pathway”.

Point 26:

Material and method section

L 267 "All pregnant swine" how many sows were used in the experiment? Provide n. Please, also provide more details about used sows - age, body mass, breed.

Response 26: Thanks for your valuable comments, pregnant sows (nulliparous) with Landrace and Yorkshire background at about 90 kg and 26 weeks of age were individually housed in the same room, and nineteen pregnant sows were successfully inseminated and used in the following experiment.

Point 27:

L 268 "artificially inseminated" when, concerning the estrus or ovarian cycle, were sows inseminated?

L 271-272 "from day 60 of gestation (G60)" How and when the pregnancy was confirmed? How many AI sows were pregnant confirmed? Was it possible to get pregnant each time after the first AI?

Response 27: Thanks for your valuable comments. All sows were checked daily for estrus with a mature boar from 26 weeks of age, the first detection of standing estrus was taken as the first estrus. Sows were intracervical inseminated by using semen from Duroc boars at the third estrus, and were inseminated every 12 h thereafter for 3 times. The first day of mating was taken as day 0 of gestation. The successfully pregnancy was further verified by ultrasound examination at 25 days after insemination. A total of nineteen pregnant sows were successfully inseminated and used in the following experiment.

Point 28:

L 272-277 It is not clear how many sows were included in each group. Do I understand correctly that you only took samples three times? On days 60, 90, and 114-116(115) of pregnancy? How long was the pregnancy of each of the pigs tested? Which day of pregnancy was day L0? Was each sampling associated with the termination of pregnancy?

Do I understand correctly that on day G60 samples were taken from 6 sows, on day G90 also from 6 sows, and day L0 from 9 sows?

Response 28:Thanks for your valuable comments, we have modified the description you raised.

A total of nineteen pregnant sows were successfully inseminated. The maternal peripheral blood was continuously sampled from seven sows at G60, G90 and L0. Thus the maternal peripheral blood was sampled three times at G60, G90 and L0. In addtion, six pregnant sows were salughter and sampled for fetal blood and placentas at G60, G90, respectively. Fetal blood and placentas were also collected at L0 after delivery.

The first day of mating was taken as day 0 of gestation. The day of farrowing day was taken as L0.

Point 29:

Explain in detail what exactly (what tissue, blood) was taken from each sow and how was it stored?

Explain in detail what exactly (what tissue, blood) was taken from each fetus and how was it stored?

How many fetuses were present in the uterus of each sow at the time of sampling. How many fetuses were sampled? From all, a dozen, a few, or one randomly assigned?

L 273 "and the farrowing" or? were the same sows used twice or three times?

Response 29:Thanks for your valuable comments, we have modified the description you raised.

The maternal peripheral blood was collected from the precaval vein of sows at G60, G90 and L0, respectively(n=7/timepoint). To obtain the placental and fetal blood samples, sows were sampled after 8-hours fasting at G60 and G90, respectively(n=6/timepoint). The fetal serum samples were acquired after centrifuge at 3000 g for 10 min and then stored at -80℃. The placental tissue samples, which surrounded the cervix and utero-tubal junction, were immediately flash frozen in liquid nitrogen and then stored at -80℃.The placental tissues and fetal blood were collected from the fetus with average body weight.

The pregannt sows were sampled twice at L0, one is for the maternal peripheral blood, the other is for the placenta and fetal blood.

Point 30:

L 275 Did 8-hours fasting not affect your BA metabolism? Were the sows also subjected to 8-hours of fasting at the time of L0 sampling? If not, can you justify the desirability of comparing the results obtained in this way?

Response 30: Thanks for your valuable comments, we have modified these description you raised.

The postprandial BA metabolism reach peak levels at about 2 hours and retured to basel levels at about 6 hours according to previous studies in human(LARusso et al.,New England Journal of Medicine 1974) and pregnant swine (unpublished data). Sows were also subjected to 8-hours of fasting before sampling at L0.

Point 31:

L 276 Detail of the laparotomy protocol including deep isoflurane-induced anesthesia should be provided.

Response 31: Briefly, atropine sulfate and sumiannin II were injected through ear vein to complete anesthesia induction. After the sows were effectively anesthetized, they were fixed on the operating table and maintained anesthetized with isoflurane through a respiratory anesthesia mask, the fetal blood of umbilical cord vein and placentas were sampled from fetuses with average body weight.

Point 32:

L 279 "2-5 day old female pigs" How many pigs were used?

Response 32: Thanks for your valuable comments, Three female pigs have been used for pig primary hepatocyte isolation and culture.

Point 33:

L 291 Please, provide the manufacture details, numbers of kits, the sensitivity, intraassay CV, and interassay CV.

L 297-299 Please, provide the manufacture details, numbers of estradiol kits, the sensitivity, intraassay CV, and interassay CV. Did you investigate 17ß-estradiol?

L 315 and 318 Please, provide the kit details.

Response 33: Thanks for your valuable comments, we have modified these description you raised.

   The TBA was measured by the same enzymatic cycling method assay kits (Kehua Bio-Engineering Co., Ltd, Shanghai, China) in Model 3100 automatic biochemical ana-lyzer (Hitachi, Tokyo, Japan), the detection range was 1-180 μmol/ml, and the sensitivity was below 5 pg/ml, the intraassay variable coefficient was lower than 5%, the interassay variable coefficient was lower than 10%.

   Estradiol(17β-estradiol) was measured by the same one estradiol radioimmunoassay KIT according to the manufacturer’s specifications (Iodine[125I] Estradiol radioimmunoassay Kit, Beijing North institute of Biotechnology, Beijing, China), the detection range was 6-1000 pg/ml, and the sensitivity was below 5 pg/ml, the intraassay variable coefficient was lower than 10%.

Point 34:

L 318 Please, provide the investigated gene expression list including used forward and reverse primer sequences.

Response 34: we have added the primer information in the supplementary Table S1.

Table S1. Primer sets for real-time RT-PCR analysis

Genes

Primer sequence(5’-3’)

Accession number

β-actin

FW AGAGCAAGAGAGGCATCCTG

XM_003124280.5

RV CACGCAGCTCGTTGTAGAAG

CYP7A1

FW GAAAGAGAGACCACATCTCGG

NM_001005352

RV GAATGGTGTTGGCTTGCGAT

CYP8B1

FW CCGGAAGAATATGTTGGAAT

NM_214426.1

RV AAGTCTAGTTTTCTCTTCGC

NTCP/SLC10A1

FW ACTTTCGGAAACCTAAGGGACT

XM_001927695.5

RV AAGAGCTTGCCCAGTGCAAAG

OSTβ/SLC51B

FW GAAATCCAAAGACGCTGCCA

XM_005658570.3

RV CCCTTAGGATGGTCAGGTTGT

BSEP/ABCB11

FW TTTCATTCAGCGCCTGACCA

XM_003133457.5

RV ACTCCAATGAGAGGGCTGAC

MRP2/ABCC2

FW TGCAAGTACGGACCAGTGTC

XM_021073710.1

RV AACGGTGTACTGCTTCCTGG

MDR3/ABCB4

FW CCAGGAAGCAAAGAAACTCAATG

XM_013989596.2

RV CTCCTCCAGGGTCACAATGC

SULT2A1

FW CCATGCGAGACAAGGAGAAC

NM_001037150.1

RV CATGACCTGGAAGGAGCTGT

FXR

FW ATACAACAGTGTTCCGTTTC

NM_001287412.1

RV AGAGTCTCAGCAGGCATT

SHP

FW GCCTACCTGAAAGGGACCAT

DQ002896

RV CAACGGGTGTCAAGCCTTTA

FGF19

FW AGTACTCGGATGAGGACTGTGCTT

XM_003122420.3

RV AGAGACGGGCAGATGGTGTTTCTT

Point 35:

L 319 In which samples/group the target gene expression was considered as ΔCt Control Value?

Response 35: Relative expression levels of the target gene were calculated using the  2−ΔΔCT method (Livak and Schmittgen, 2001) and normalized to β-actin (reference gene).The ΔCt was calculated by the cycle threshold (CT) value of targeted gene minus CT value of the inetrnal control (β-actin). As for the experiment with PM4S or PM5S treated alone, the zero levels of PM4S or PM5S was taken as the control group. As for the FXR activator (CDCA) and PM5S co-treat experiment, the treatment without both CDCA and PM5S was taken as the control group.

Point 36:

L 334 Excepting PROC GLM, please provide the test used for univariate distributions testing. In the compared data series, has always at least one data series shown a non–Gaussian distribution? Please, describe in detail what you compared with what and how, because based on the presented description I am not able to judge whether the statistical analysis was carried out correctly.

Response 36: The Shapiro-Wilk was used for test the univariate distributions. Sorry for the trouble takes to you, all data in this experiment were normally distrubuted, and we have modified the inappropriate describption shown in the follwing:

  All data were analyzed by PROC GLM, the Shapiro-Wilk was used to test the univariate distributions. The least significant difference test was used to compare the group means when the F test in the analysis of variance table was significant.

Point 37:

Both in the abstract and for the aim of the study sections, you indicated that the correlation between the data series was examined, while in the place appropriate for this description there is no data about carrying out the relevant calculations.

Response 37: Thanks for your valuable comments. we have added the statistical method in the data analysis section. Correlation analysis was carried out using the CORR procedure of SAS to study the relationship between PMSs and BA levels in both maternal and fetal serum at G60,G90 and L0. And their relationship have been shown in figure S1A,S1B, 2c, 2d, 4c,4d.

Reviewer 3 Report

Excellent work with very interesting conclusions. In Figure 1, in sections (c and d), "etradiol" should be changed to ESTRADIOL.

Author Response

Point 1: Excellent work with very interesting conclusions. In Figure 1, in sections (c and d), "etradiol" should be changed to ESTRADIOL.

Response 1: Thanks for your valuable comment, we have changed the “etradiol” to “estradiol” in manuscripts according to your suggestion.

Reviewer 4 Report

This work is a continuation of the same group work on the correlation of reproductive hormones and bile acid metabolism.

There are some minor comments to be considered during the revision.

1- The core abbreviation (TBA) is not described in the text. It is mentioned in the methods. Please mention it in the first appearance.

2- L319: correct the equation into (2−ΔΔCT method).

3- In methods: authors referred to Ref 28 for the primers’ details, but I cannot find the sequences of most of the used transcripts in the current study. Please double-check the primers and submit the sequences as a supplementary table.

Author Response

Point 1: This work is a continuation of the same group work on the correlation of reproductive hormones and bile acid metabolism.

There are some minor comments to be considered during the revision.

1- The core abbreviation (TBA) is not described in the text. It is mentioned in the methods. Please mention it in the first appearance.

Response 1: Thanks for your valuable comment, I have added the abbreviation of TBA in the first appearance in the manuscripts according to your suggestion.

Point 1: 2- L319: correct the equation into (2−ΔΔCT method).

Response 2: Thanks for your valuable comment, I have change equation from “2−ΔΔCT method” to “2−ΔΔCT method” in the manuscripts according to your suggestion.

Point 2: 3- In methods: authors referred to Ref 28 for the primers’ details, but I cannot find the sequences of most of the used transcripts in the current study. Please double-check the primers and submit the sequences as a supplementary table.

Response 2: Thanks for your valuable comment, part of primer such as SULT2A1 was referred from ref 28, and we have added the primers sets in the supplementary table S1 according to your suggestion. The primer set are shown as follows:

Table S1. Primer sets for real-time RT-PCR analysis

Genes

Primer sequence(5’-3’)

Accession number

β-actin

FW AGAGCAAGAGAGGCATCCTG

XM_003124280.5

RV CACGCAGCTCGTTGTAGAAG

CYP7A1

FW GAAAGAGAGACCACATCTCGG

NM_001005352

RV GAATGGTGTTGGCTTGCGAT

CYP8B1

FW CCGGAAGAATATGTTGGAAT

NM_214426.1

RV AAGTCTAGTTTTCTCTTCGC

NTCP/SLC10A1

FW ACTTTCGGAAACCTAAGGGACT

XM_001927695.5

RV AAGAGCTTGCCCAGTGCAAAG

OSTβ/SLC51B

FW GAAATCCAAAGACGCTGCCA

XM_005658570.3

RV CCCTTAGGATGGTCAGGTTGT

BSEP/ABCB11

FW TTTCATTCAGCGCCTGACCA

XM_003133457.5

RV ACTCCAATGAGAGGGCTGAC

MRP2/ABCC2

FW TGCAAGTACGGACCAGTGTC

XM_021073710.1

RV AACGGTGTACTGCTTCCTGG

MDR3/ABCB4

FW CCAGGAAGCAAAGAAACTCAATG

XM_013989596.2

RV CTCCTCCAGGGTCACAATGC

SULT2A1

FW CCATGCGAGACAAGGAGAAC

NM_001037150.1

RV CATGACCTGGAAGGAGCTGT

FXR

FW ATACAACAGTGTTCCGTTTC

NM_001287412.1

RV AGAGTCTCAGCAGGCATT

SHP

FW GCCTACCTGAAAGGGACCAT

DQ002896

RV CAACGGGTGTCAAGCCTTTA

FGF19

FW AGTACTCGGATGAGGACTGTGCTT

XM_003122420.3

RV AGAGACGGGCAGATGGTGTTTCTT

Round 2

Reviewer 1 Report

The authors have revised most of the suggested points, and I believe the manuscript has improved. Nevertheless, there are some points to review.

In the M&M section, it is written that a total of 72 species BA in serum were measured. However, they say that it was not possible to measure the BA profile in their answer. It is a little contradictory.

I still think that a method to reduce error type 1 is needed to evaluate the significance of the results.

In Data analysis, it should be indicated that correlations were analyzed with the 3 time points together. The wording is confusing.

I can't open the supplementary information.

Author Response

Thanks for your careful guidance and help, which help improving our current and further research.

Point 1

The authors have revised most of the suggested points, and I believe the manuscript has improved. Nevertheless, there are some points to review.

In the M&M section, it is written that a total of 72 species BA in serum were measured. However, they say that it was not possible to measure the BA profile in their answer. It is a little contradictory.

Response 1: Please accept our apology for the trouble taken to you. Frankly, the method of BA analysis was described referring to our previous research proposal, but finally we failed to analyze as planned in the proposal the 72 species BA due to technique difficulty. We are sorry for forgetting modifying it in the last revision and it was deleted accordingly in the present manuscript.

Point 2

I still think that a method to reduce error type 1 is needed to evaluate the significance of the results.

Response 2: Thanks for your valuable comment. All the data were reanalyzed. In particular, the Student-Newman-Keuls (SNK) test instead of the least significant difference (LSD) test was used to compare means of the treatment groups, thus reducing error type 1. Accordingly, the description was corrected as follows: The relative expression mRNA levels between CON and CDCA were analyzed by Student’s T-test of GraphPad Prism (8.0.2 version). Data passing the normality test were analyzed by one-way ANOVA of GraphPad Prism and means were compared using the Student-Newman-Keuls test. The symbols marked on group means to indicate differences were corrected accordingly. Importantly, these corrections didn’t impact the conclusion of this study.

Point 3

In Data analysis, it should be indicated that correlations were analyzed with the 3 time points together. The wording is confusing.

Response 3: thanks for your valuable comment, we have modified the “Correlation analysis was carried out using the CORR procedure of SAS to study the relationship between PMSs and BA levels in both maternal and fetal serum atG60, G90 and L0” to “Correlation analysis was carried out using the CORR procedure of SAS to study the relationship between PMSs and BA levels in both maternal and fetal serum at three timepoints (G60, G90 and L0) together”.

Point 4

I can't open the supplementary information.

Response 4: I’m very sorry for the trouble takes to you, we have re-upload the supplementary information, supplementary data can be download from the link shown below https://doi.org/10.6084/m9.figshare.19914805.v8

https://doi.org/10.6084/m9.figshare.19914790.v2

Thanks again for your kind help!

Reviewer 2 Report

I appreciate the effort to improve this article and the detailed answer to all the ambiguities I have raised. In my opinion, the revised version of the article is significantly improved.

Please, see the detailed minor comments:

L43, L277, and L423 change "promising ideal model" to "promising model" or "promising animal's model"

L51 change "organ system" to "organism"

L65 change "estradiol,estradiol-17β-glucuronide" to "estradio (estradiol-17β-glucuronide)"

L120 In the material and method section, there is no G75 and G105 sampling. Please, clarified.

L123-142 add unit to the relative expression level (I suspect AU

L270-271 Was the antagonist role of PM4S on FXR-mediated BA homeostasis recently demonstrated in other's species hepatocytes? The obtained results should be discussed in detail against all previously available reports.

L 276-277 I can not see the link between the sentence in L276 and L277. Probably some additional explanation is required. 

it is not comprehensive. Please consider expanding this to discuss your estradiol findings.

L316 change "about 90 kg" to "90±... kg"

L322 the details of the ultrasound examination should be provided. Was the examination transabdominal or transvaginal? Provide details of probe and ultrasound scanner including manufacture, city, and country.

L323 "lighting schedule was provided from 0800 to 1800h" check if ":" is not missing.

Author Response

Thanks for your careful guidance and help, which help improving our current and further research.

Point 1

I appreciate the effort to improve this article and the detailed answer to all the ambiguities I have raised. In my opinion, the revised version of the article is significantly improved.

Please, see the detailed minor comments:

L43, L277, and L423 change "promising ideal model" to "promising model" or "promising animal's model"

Response 1: Thanks for your suggestion, we have changed the “promising ideal model” to “promising animal's model”.

Point 2

L51 change "organ system" to "organism"

Response 2: Thanks for your suggestion, we have changed the “organ system” to “organism”.

Point 3

L65 change "estradiol,estradiol-17β-glucuronide" to "estradio (estradiol-17β-glucuronide)"

Response 3: Thanks for your suggestion, we have changed the “estradiol,estradiol-17β-glucuronide” to “estradio (estradiol-17β-glucuronide)”.

Point 4

L120 In the material and method section, there is no G75 and G105 sampling. Please, clarified.

Response 4: Thanks for your suggestion, we have added the sampling time at day 75 of gestation (G75) and day 105 of gestation (G105) in the material and method section. In addition, we also added the information in the figure legend of figure 1.

Point 5

L123-142 add unit to the relative expression level (I suspect AU

Response 5: Thanks for your valuable comment, figure 3 showed the mRNA expression levels relative to internal control (β-actin), and it commonly don’t have the unit.

Point 6

L270-271 Was the antagonist role of PM4S on FXR-mediated BA homeostasis recently demonstrated in other's species hepatocytes? The obtained results should be discussed in detail against all previously available reports.

Response 6: Thanks for your valuable comment. As far as we know, previous studies have demonstrated the inhibition role of PM4S on bile acids absorption in primary human hepatocytes (Abuhayyeh et al., Journal of Biological Chemistry 2010) and bile acids output in rat liver (Vallejo et al., Journal of hepatology 2006), but the detailed mechanism was still unknown. We have added the information in the discussion section.

Point 7

L 276-277 I can not see the link between the sentence in L276 and L277. Probably some additional explanation is required. it is not comprehensive. Please consider expanding this to discuss your estradiol findings.

Response 7: Thanks for your valuable comment. We have added the sentence “As the main downstream of FXR, FGF15 (ortholog of FGF19 in human and pigs) regulating hepatic BA metabolism in mice was mainly through intestinal FGF15-FGFR4 pathway [34,35], whereas FGF19 was also expressed in human primary hepatocytes and regulated BA synthesis through CYP7A1[36]. The high expression of FGF19 in pig hepatocytes in this study indicated pregnant sow is a promising animal's model to investigate the pathogenesis of cholestasis during pregnancy” in the discussion section.

Point 8

L316 change "about 90 kg" to "90±... kg"

Response 8: Thanks for your valuable comment, we have added the exact body weight (90.0 ± 1.7 kg) in the material and methods section.

Point 9

L322 the details of the ultrasound examination should be provided. Was the examination transabdominal or transvaginal? Provide details of probe and ultrasound scanner including manufacture, city, and country.

Response 9: Thanks for your valuable comment, we have added the information in the material and methods section shown as below:

The ultrasound examinations were made with all-digital ultrasound diagnostic system (WED-180, Shenzhen Well.D Medical Electronics Co., Ltd, Shenzhen, China), which equipped with abdominal convex probe (3.5MHz,), the maximum display depth was 250mm. The ultra-sound examination was performed at the right abdominal wall and near the caudal three mammary gland.

Point 10

L323 "lighting schedule was provided from 0800 to 1800h" check if ":" is not missing.

Response 10: Thanks for your suggestion, we have changed the “lighting schedule was provided from 0800 to 1800h” to “lighting schedule was provided from 08:00 to 18:00” to make it more accurate.

Thanks again for your kind help!